# Digital Citizenship and Life Satisfaction in South Korean Adolescents: The Moderated Mediation Effect of Poverty

**DOI:** 10.3390/children10060973

**Published:** 2023-05-30

**Authors:** Ju-Young Lee, Gyungjoo Lee, Il Hyun Lee, Won Hee Jun, Keelyong Lee

**Affiliations:** 1College of Nursing, The Catholic University of Korea, 222 Banpo-daero, Seocho-gu, Seoul 06591, Republic of Korea; 2StatEdu Institute of Statistics 514, Knowledge Industry Center 174, Yakchon-ro, Iksan-si 54630, Republic of Korea; 3College of Nursing, Keimyung University, 1905 Dalgubeol-daero, Dalseogu, Daegu 42601, Republic of Korea; 4Department of Nursing, Suwon Science College, 288, Seja-ro, Jeongnam-myeon, Hwaseong-si 18516, Republic of Korea

**Keywords:** adolescents, citizenship, life satisfaction, poverty, self-efficacy

## Abstract

This study examined the moderated mediation effect of poverty on the paths between enactive mastery experience in digital life and life satisfaction mediated by digital citizenship and digital life among Korean adolescents using structural equation modelling. This cross-sectional study involved a secondary data analysis of 2020 national data in The Report on the Digital Divide provided by the National Information Society Agency (NIA) of Korea. Data from 1084 Korean adolescents were analyzed using IBM SPSS Statistics for Windows, version 26.0 and SPSS PROCESS macro. The results demonstrated a significant moderated mediation effect of poverty. Enactive mastery experience, which encompasses the self-knowledge, perceived task difficulty, and contextual factors of adolescents living in poverty, was associated with digital life and life satisfaction through the mediation of digital citizenship. For adolescents living in poverty, in contrast to their non-poor counterparts, enactive mastery experience in digital life and digital citizenship are two critical factors in life satisfaction. Therefore, institutional support enabling adolescents and their communities to forge partnerships is necessary to foster these two factors.

## 1. Introduction

In this digital era, the use of digital products and services is associated with people’s life satisfaction [1]. Adolescents in modern society are digital natives whose digital life has a substantial impact on most aspects of their lives, including education, peer relationships, and hobbies [2]. The life satisfaction of adolescents is particularly important as adolescence is a time of life that influences the overall health and development of an individual [3]. However, poverty is closely related to digital exclusion [4], and adolescents in impoverished families living in an environment with more disadvantages than their non-poor counterparts show low life satisfaction [3].

Abundant previous studies have argued that even in a disadvantaged environment, life satisfaction can be altered by an individual’s overall cognitive appraisal of life, depending on not only one’s environmental characteristics but on personal intrinsic and extrinsic resources [5,6]. Self-efficacy is a key factor that can enhance the quality of life of adolescents living in poverty [3]. Bandura (1997) [7] defined self-efficacy as “beliefs in one’s capabilities to organize and execute the courses of action required to produce given attainments”. Bandura further noted that enactive mastery experience is the most powerful source of influence in the formation of self-efficacy. In particular, enactive mastery experience depends on task difficulty, self-knowledge, and contextual factors (including suitability of support from others and adequacy of available resources), which constitute the root of any change in self-efficacy.

Enactive mastery experience by task difficulty, self-knowledge, and contextual factors in digital life may increase the life satisfaction of adolescents. Digital competence has been noted as an important survival technique for people living in the digital era as well as an essential component of education and learning [8,9]. A low level of belief in digital competence leads to low academic achievement and work performance [10]. Moreover, perceived ease of use regarding digital technology has influenced the adaptation to online education. For example, a decrease in task difficulty can increase the attitude toward the use of the online test platform and the intention to use an online test [11,12]. In addition, a lot of extrinsic resources are required to make digital life smooth. To illustrate, for effective digital learning, preparation by teachers, school, and home environments and accessibility for students should all be appropriate [13]. Therefore, in order to devise strategies to enhance the quality of life of adolescents in poverty, it is necessary to understand perceptions around the knowledge of digital competence, the level of awareness of the difficulty of a digital task, and the presence or absence of support from others or available resources on the digital life of adolescents.

To address psychobehavioral problems, including smartphone and game addiction and cyberbullying, both educational institutions and research institutions have emphasized the technological drawbacks and misuse of technology by users [14]. Nonetheless, Walters et al. (2019) [15] highlighted the importance of digital citizenship. Digital citizenship is defined as the online code of conduct for the safe, ethical, and responsible use of technology with the capacity to develop the necessary technology and perspectives toward a digital lifestyle [16,17]. In a study on cyberbullying, Zhong et al. [18] noted that among the factors of digital citizenship, internet etiquette and understanding of and compliance with cyber laws and regulations are negatively correlated with cyberbullying. Recent research has likewise emphasized the importance of digital citizenship as a factor to prevent the side effects of digital use [19]. The critical role of digital citizenship is likely to apply consistently to adolescents in poverty.

The increased life satisfaction of adolescents is associated with positive adaptation in adulthood [20]. If adolescents living in this digital era can acquire confidence in digital use and own their digital citizenship to adapt positively to digital life with healthy psychosocial functions, then their life satisfaction will improve, including that of adolescents living in poverty. Thus, elucidating the life satisfaction of adolescents living in poverty is especially crucial. Therefore, this study aimed to identify the moderated mediation effect of poverty on the paths between enactive mastery experience in digital life and life satisfaction mediated by digital citizenship and digital life among Korean adolescents based on the theory of self-efficacy.

## 2. Methods

### 2.1. Study Design

Our study used secondary data from the Report on the Digital Divide [21] collected by the National Information Society Agency (NIA) of Korea to determine the moderated mediation effect of poverty on the paths between enactive mastery experience in digital life and life satisfaction mediated by digital citizenship and digital life among Korean adolescents. For this, we devised and tested a hypothetical model (Figure 1).

### 2.2. Data Collection

The Report on the Digital Divide [21] includes the population representing the non-poor class, in addition to the population representing a special group of older adults, the poor class, persons with disabilities, and North Korean refugees. The participants in the Report on the Digital Divide [21] were 7000 persons in the non-poor class, 2200 persons in the poor class, 2200 persons with disabilities, 2200 persons working as agricultural or fishery farmers, 700 North Korean refugees, and 700 immigrants married to Koreans; based on the data of a total of 15,000 people, the level and reality of the digital information gap of each group were investigated. For the survey, the proportional quota sampling method was applied to the sample by gender, age, and regional local government, and face to face interviews were conducted, during which the participants completed a structured questionnaire. The NIA’s study period was from September to December 2020.

In the present study, we excluded data with missing values on the items of the main variables. Thus, from the data of 1085 middle school to high school adolescents in the non-poor and poor classes, who comprised our target sample, our analysis used the data of 1084 adolescents, including 648 and 436 middle to high school students in the non-poor and poor classes, respectively. The non-poor class refers to household members living in households across the nation. The poor class indicates the registered beneficiaries of National Basic Livelihood based on the National Basic Living Security Act of Korea.

### 2.3. Variables

#### 2.3.1. Enactive Mastery Experience

##### Perceived Task Difficulty

We measured perceived task difficulty based on self-assessment by adolescents regarding their ability to perform activities using mobile devices (e.g., smartphones, tablets). The assessment included seven questions as follows: “I can change device settings, such as display, sound, security, alarm, input, etc.”, “I can set up a wireless network (Wi-Fi)”, “I can move a file from a mobile device to a computer”, “I can send files and photos on my mobile device to another person’s device”, “I can install/remove/update an app in mobile devices”, “I can test/treat malware (e.g., virus, spyware) on mobile devices”, and “I can write texts or create data (e.g., memos, documents) on mobile devices”. The participants rated each question on a scale from 1 = Strongly disagree to 4 = Strongly agree, with higher scores indicating higher levels of perceived task difficulty. Cronbach’s α was 0.871 in our study.

##### Self-Knowledge

We measured self-knowledge based on the self-assessment by adolescents regarding their confidence and attitude when encountering new technology (four questions) and personal efforts to acquire new technology (two questions). The questions on confidence and attitude were “I tend to adapt easily to new technology or products”, “I am confident in learning how to use new technology or products on my own”, “I tend to use new technology or products more efficiently than others”, and “I think the ability to use digital devices is critical to continuous economic activities in the future”. The questions on personal efforts were “I am motivated to actively acquire new technology” and “I consider myself a lifelong learner and enjoy taking necessary courses”. The participants rated each of the six questions on a scale from 1 = Strongly disagree to 4 = Strongly agree, with higher scores indicating higher competence and efforts by the adolescents. Cronbach’s α was 0.738 in our study.

##### Contextual Factors

We measured the contextual factors that underly problem-solving in the use of digital devices based on the adolescents’ self-assessment of how they try to solve problems related to mobile devices (e.g., smartphones, tablets). The assessment consisted of the following six questions: “I solve the problem by myself without help from others”, “I get help from family (e.g., siblings, parents, nephews, and nieces)”, “I get help from a friend”, “I get help from a classmate or someone I know”, “I search information on the internet”, and “I seek professional help at service centers, etc.”. Each question was rated on a scale from 1 = Strongly disagree to 4 = Strongly agree, with higher scores indicating more effective contextual factors. Cronbach’s α was 0.699 in our study.

#### 2.3.2. Digital Citizenship

Regarding digital citizenship, our measurement used the self-assessment data of adolescents regarding their participation in activities and interactions in the digital world, information security, and respect for others using the following four questions: “I can connect and communicate with others on the internet and cooperate with others for problem-solving, completing tasks, and assignments”, “I can use the internet to actively exchange opinions on political or social concerns and issues and participate in various activities, such as discussions, donations, and volunteer works to solve public problems”, “I can protect myself and others from the various risks of internet use, such as disclosure of personal information or information of others”, and “I can understand, acknowledge, and accept differences in opinions through a responsible use of the internet that does not involve illegal media or violates the rights of others”. Each question was rated on a scale from 1 = Strongly disagree to 4 = Strongly agree, with higher scores indicating higher levels of digital citizenship in adolescents. Cronbach’s α was 0.739 in our study.

#### 2.3.3. Digital Life

As for digital life, we measured it based on the self-assessment of adolescents regarding their level of use of data services on mobile devices (e.g., smartphones, tablets) in the past year. The assessment included the following four items: “Search for information and news”, “E-mail”, “Media contents (e.g., movies, music, e-books)”, and “Education contents (e.g., various online lectures and courses)”. The participants rated each item on a scale from 1 = Never to 4 = Frequently, with higher scores indicating more frequent engagement in digital life. Cronbach’s was 0.815 in our study.

#### 2.3.4. Life Satisfaction

Lastly, we measured life satisfaction based on the self-assessment of adolescents using the following five questions: “My life is close to my ideals in most cases”, “My life is based on a set of excellent conditions”, “I am satisfied with my life”, “I have been able to acquire the important things I desire in my life”, and “If I were to live another life, I would change almost nothing from the present life”. They rated each question on a scale from 1 = Completely dissatisfied to 4 = Completely satisfied, with higher scores indicating higher life satisfaction. Cronbach’s α was 0.859 in our study.

#### 2.3.5. Demographic Characteristics

Our study considered the following demographic variables: population, gender, education, household type, income, and region. The subcategories were non-poor and poor classes for population; male and female for gender; middle and high school for education; detached house, apartment, and townhouse for household type; <1500 USD, ≥1500 to <2300 USD, ≥2300 to <3000 USD, and ≥3000 USD for income; and urban and rural.

### 2.4. Ethical Considerations

As this study used secondary data that did not contain personal identification data of the participants, it was exempted from review by the institutional review board at the university of the authors and subsequently approved (IRB No. MC23ZASI0011). All files and data were discarded at the completion of the study.

### 2.5. Data Analysis

We used IBM SPSS Statistics for Windows, version 26.0, for data analysis. To analyze the moderated mediation effect, we used the PROCESS macro (4.2 version) developed by Hayes [22]. For demographic characteristics, we conducted a frequency analysis to estimate frequency and percentage values. In analyzing the reliability of each instrument, we used the values of Cronbach’s alpha. Descriptive statistics with mean and standard deviation were obtained for each instrument. Skewness and kurtosis were used to test the normality.

To analyze the differences according to demographic characteristics, we conducted an independent *t*-test and one-way ANOVA with Scheffé’s post hoc analysis. To identify the correlations across instruments, we used Pearson’s correlation analysis. We used bootstrapping for 50,000 samplings (number of bootstrap samples) to determine the moderating effect of poverty on the mediation effect of digital citizenship and digital life regarding the effects of perceived task difficulty, self-knowledge, and contextual factors on life satisfaction. Using the Model 88 process macro, the moderated mediation effect was verified. The significance level was set at 0.05. Among the demographic characteristics, those with a significant effect on life satisfaction, i.e., population, income, and region, were set as control variables for the analyses.

## 3. Results

### 3.1. Differences in Main Variables According to Demographic Characteristics

The participants’ demographic characteristics and mean scores for life satisfaction by demographic are listed in Table 1.

### 3.2. Descriptive Statistics and Correlations across Variables

For the main variables, the absolute skewness and kurtosis were 3 and 7, respectively, which satisfied the normality assumption (Table 2). Across the variables, we found a significant positive correlation between life satisfaction and self-knowledge (r = 0.205, *p* < 0.001), contextual factors (r = 0.090, *p* < 0.01), digital citizenship (r = 0.130, *p* < 0.001), and digital life (r = 0.082, *p* < 0.01).

### 3.3. Moderated Mediation Effect of Poverty on Life Satisfaction

Table 3 presents the result of testing the moderated mediation effect of poverty after controlling for the variables of demographic characteristics with a significant effect on life satisfaction using the Model 88 process macro via bootstrapping. Figure 2 illustrates the path diagram of the variables.

For the path in which perceived task difficulty (boot B = −0.015, 95% CI: −0.074~0.042), self-knowledge (boot B = −0.006, 95% CI: −0.033~0.020), and contextual factors (boot B = 0.005, 95% CI: −0.028~0.016) mediate digital citizenship and affect life satisfaction, the moderated mediation effect of poverty was not significant. The pathways by which perceived task difficulty (boot B = 0.023, 95% CI: 0.006~0.047), self-knowledge (boot B = 0.003, 95% CI: 0.010~0.064), and contextual factors (boot B = 0.031, 95% CI: 0.010~0.056) mediate digital life to influence life satisfaction showed significant moderated mediating effects of poverty.

The dual mediation for digital citizenship and digital life came from perceived task difficulty (boot B = 0.025, 95% CI: 0.010~0.049), self-knowledge (boot B = 0.012, 95% CI: 0.004~0.023), and contextual factors (boot B = 0.010, 95% CI: 0.003~0.020), and the path with an effect on life satisfaction exhibited a significant moderated mediation effect of poverty. Each path of dual mediation was significant for the poor class and was not significant for the non-poor class.

## 4. Discussion

We analyzed the 2020 data of the Report on the Digital Divide [21] collected by the NIA to verify the moderated mediation effect of poverty in the structural model of life satisfaction mediated by digital citizenship and digital life under the influence of perceived task difficulty, self-knowledge, and contextual factors. Our results demonstrated a significant moderated mediation effect of poverty, whereas the main path of mediation in the structural equation was significant solely for adolescents living in poverty. Hence, the enactive mastery experience of adolescents living in poverty, which includes perceived task difficulty, self-knowledge, and contextual factors, was shown to be associated with life satisfaction and cyber-wellness in digital life through the mediation of digital citizenship.

Our finding on the role of enactive mastery experience in increasing life satisfaction and cyber-wellness in the digital life of adolescents living in poverty coincided with those of a previous study reporting that digital use is a crucial channel to increasing the life satisfaction of low-income individuals [1]. Choices in an individual’s life are considerably limited by financial difficulties, and uncertainties over one’s present and future prospects, as well as those of one’s family, generate repetitive daily hardships. Holmes and Burgess (2022) [4] argued that the opportunities for online access using fast but costly data services are limited in persons with very low incomes. In the case of limited online access, daily life is affected as well, and adolescents may experience problems in online learning or peer relations on the internet. Marum et al. (2014) [5] showed that the sense of enactive mastery has a moderating effect on the relationship between financial difficulties and life satisfaction. The limitations engendered by poverty could reduce the feeling of enactive mastery, thereby suppressing overall life satisfaction. According to Bandura (1997) [7], enactive mastery experience needs to be increased in order for individuals to acquire self-efficacy. Our results highlight that positive belief in self-competence and perception of the difficulty of imminent tasks (as intrinsic resources) along with contextual factors (as extrinsic resources) could be critical factors in enhancing self-efficacy related to digital life and the quality of life of adolescents living in poverty. Similarly, previous studies have shown that self-efficacy and social support increase the life satisfaction of adolescents [3,6]. Thus, institutional support is needed to increase self-efficacy related to digital life for adolescents living in poverty.

In addition, we confirmed the mediatory role of digital citizenship in the structural model of life satisfaction of adolescents living in poverty. Digital citizenship is composed of digital access, trade, communication, literacy, etiquette, laws, rights and responsibilities, health and well-being, and security [23]. In a digital space, digital citizenship enables one to judge what is right and what is wrong. For adolescents living in poverty, the opportunities to acquire digital citizenship diminish as the experience of digital exclusion increases. Lim et al. (2016) [24] highlighted the need for partnerships between adolescents and their communities for cultivating digital citizenship. They claimed that an important principle in the digital space is “Respect for Self and Others” and “Safe and Responsible Use”, and for this, adolescents are required to foster the ability of self-control to behave in a responsible manner in the digital world, whereas communities, including parents, teachers, and local governments, should provide the education and perform necessary monitoring. A previous study revealed that digital citizenship is an ability that can be internalized through training [16]. The digital citizenship of adolescents living in poverty can be cultivated through collective efforts of the individual, family, educational institution, and government.

The limitations of this study were as follows. As a cross-sectional study, our work was limited in the interpretation of cause–effect relations across variables. In addition, only the variables found in the raw data were applied in constructing the model, which may limit the validity of the model. The 2020 data of the Report on the Digital Divide [21] did not focus on the digital life and life satisfaction of adolescents, which limits the in-depth interpretation of results. Further studies should investigate a wider scope of resources that hold significance in the digital life of adolescents, in addition to the variables examined in our study. Despite these limitations, the present study is significant in analyzing a representative dataset to verify the critical effects of enactive mastery experience in digital life and digital citizenship on the life satisfaction of adolescents living in poverty as compared with their non-poor counterparts. An important direction for future research is to develop and evaluate interventions that can help young people from low-income families have a successful mastery experience in digital life and improve their digital citizenship, taking into account the home, school, and community levels.

## 5. Conclusions

Enactive mastery experience in digital life and digital citizenship are two critical factors in the life satisfaction of adolescents living in poverty, in contrast to their non-poor counterparts. Our findings highlighted the importance of an enactive mastery experience associated with digital competence with respect to enhancing cyber-wellness in the digital life and life satisfaction of adolescents living in poverty. Furthermore, digital citizenship played a central role in the life satisfaction of adolescents living in poverty. Thus, institutional support is necessary to enable adolescents living in poverty and their communities to forge partnerships toward the improvement of self-efficacy related to digital life and digital citizenship.

## Figures and Tables

**Figure 1 children-10-00973-f001:**
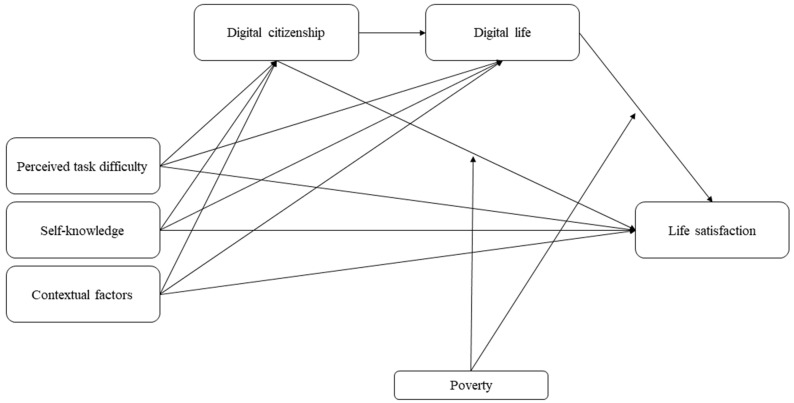
Conceptual framework of the study.

**Figure 2 children-10-00973-f002:**
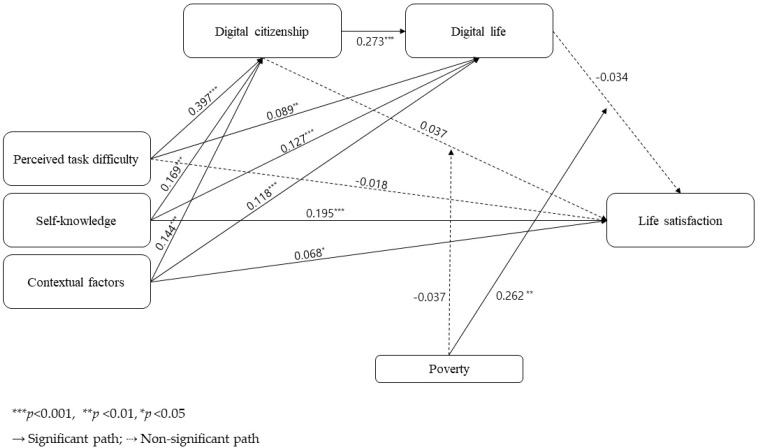
Path diagram of variables.

**Table 1 children-10-00973-t001:** Mean scores for life satisfaction by demographic characteristics. (*n* = 1084).

Characteristics		*n* (%)	Life Satisfaction	t or F (p) Scheffé
			M ± SD	
Population	Middle or upper class	648 (59.8)	2.74 ± 0.54	9.904 (<0.001 ^†^)
	Poor class	436 (40.2)	2.37 ± 0.65	
Gender	Male	559 (51.6)	2.59 ± 0.62	0.052 (0.958)
	Female	525 (48.4)	2.59 ± 0.60	
Education	Middle school	177 (16.3)	2.53 ± 0.62	
	High school	907 (83.7)	2.61 ± 0.61	
Household type	Detached house	281 (26.0)	2.57 ± 0.57	0.547 (0.579)
	Apartment	566 (52.4)	2.59 ± 0.65	
	Townhouse	234 (21.6)	2.63 ± 0.57	
Income	<1500 USD ^a^	332 (30.6)	2.33 ± 0.66	34.638 (<0.001 ^†^)
	≥1500 to <2300 USD ^b^	143 (13.2)	2.55 ± 0.55	^c,d > b >a^
	≥2300 to <3000 USD ^c^	204 (18.8)	2.71 ± 0.55	
	≥3000 USD ^d^	405 (37.4)	2.77 ± 0.54	
Region	Urban	918 (84.7)	2.63 ± 0.61	4.474 (<0.001)
	Rural	166 (15.3)	2.40 ± 0.60	

^†^: Not assuming equal variance; USD, United States Dollar; ^a^: Income < 1500 USD, ^b^: Income ≥ 1500 to <2300 USD, ^c^: Income ≥ 2300 to <3000 USD, ^d^: Income ≥ 3000 USD.

**Table 2 children-10-00973-t002:** Descriptive statistics and correlation among variables.

	X1	X2	X3	M1	M2	Y
X1. Perceived task difficulty	1.000					
X2. Self-knowledge	0.364 ***	1.000				
X3. Contextual factors	0.108 ***	0.142 ***	1.000			
M1. Digital citizenship	0.413 ***	0.304 ***	0.206 ***	1.000		
M2. Digital life	0.289 ***	0.260 ***	0.238 ***	0.430 ***	1.000	
Y. Life satisfaction	0.043	0.205 ***	0.090 **	0.130 ***	0.082 **	1.000
M ± SD	3.31 ± 0.52	3.04 ± 0.46	2.72 ± 0.54	2.80 ± 0.58	2.34 ± 0.49	2.59 ± 0.61
Skewness	−0.865	−0.646	−0.287	−0.321	−0.035	−0.277
Kurtosis	1.149	1.144	0.038	0.382	−0.154	−0.307

*** *p* < 0.001, ** *p* < 0.01.

**Table 3 children-10-00973-t003:** Moderated mediation effect of poverty on the model of life satisfaction.

Independent Variable	Indirect Effect	Moderator	Boot B	Boot 95% CI	Moderated Mediation		
					Boot B	Boot 95% CI	
Perceived task difficulty (X1)	X1 → M1 → Y	non-poor	0.015	−0.022~0.054	−0.015	−0.074	−0.042
		poor	0.000	−0.049~0.046			
Self-knowledge (X2)	X2 → M1 → Y	non-poor	0.006	−0.011~0.025	−0.006	−0.033	−0.020
		poor	0.000	−0.021~0.022			
Contextual factors (X3)	X3 → M1 → Y	non-poor	0.005	−0.009~0.020	0.005	−0.028	−0.016
		poor	0.000	−0.019~0.018			
Perceived task difficulty (X1)	X1 → M2 → Y	non-poor	−0.003	−0.013~0.005	0.023	0.006	−0.047
		poor	0.020	0.005~0.039			
Self-knowledge (X2)	X2 → M2 → Y	non-poor	−0.004	0.018~0.008	0.003	0.010	−0.064
		poor	0.029	0.009~0.054			
Contextual factors (X3)	X3 → M2 → Y	non-poor	−0.004	−0.015~0.008	0.031	0.010	−0.056
		poor	0.027	0.009~0.049			
Perceived task difficulty (X1)	X1 → M1 → M2 → Y	non-poor	−0.004	−0.014~0.006	0.025	0.010	−0.049
		poor	0.025	0.009~0.042			
Self-knowledge (X2)	X2 → M1 → M2 → Y	non-poor	−0.002	−0.007~0.003	0.012	0.004	−0.023
		poor	0.010	0.003~0.020			
Contextual factors (X3)	X3 → M1 → M2 → Y	non-poor	−0.001	−0.005~0.002	0.010	0.003	−0.020
		poor	0.009	0.003~0.017			

M1: Digital citizenship, M2: Digital life, Y: Life satisfaction.

## Data Availability

The data described in this article are openly available at https://www.data.go.kr/data/15038422/fileData.do (accessed on 1 March 2022.).

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
