# Peer review of "Digital Citizenship and Life Satisfaction in South Korean Adolescents: The Moderated Mediation Effect of Poverty"

_children, 2023, doi:10.3390/children10060973_

Round 1

Reviewer 1 Report

The study examines the moderated mediation effect of poverty on digital life and life satisfaction among Korean adolescents using structural equation modelling. The study lacks in theoretical underpinnings and formalizing the conceptual model. So the authors need to strengthen their background before analyzing the formal model.  My specific comments are as below:

1.       I am not very convince with what the authors trying to do in the study. Predictors of life satisfaction among adolescents? The motivation of the study needs to be more focused and highlighted in the introduction section.

2.       What is the age range of the sample? Is it only >=7 years? What is the maximum age? Any effect will vary across age groups and occupation levels.

3.       The conceptual model in my view is not backed by theory and also not interpreted well. For example, poverty is only considered as a moderator. But, poverty will also impact digital citizenship.

N/A

Reviewer 2 Report

This very interesting article presents an important actual issue. The authors give a clear background and chose an appropriate model for analyzing their hypothesis. They also specified clearly their methods and gave a clear definition of the main variables.

However, as the main objective of the article was to test the complex hypothesis as presented by their model, I would expect to have more details, both in the methodology (regarding the parameters) and in the results.

Although I am not an expert in the Hayes PROCESS methodology, though I used it in some of my studies, I was confused by reading the results. The model presented in the article test the mediation effect of digital citizenship and digital life and the moderation effect of poverty on the association between 3 independent variables (Perceived task difficult, self-knowledge and Contextual factors) and life satisfaction. Table 3 present it clearly and it is written properly at the beginning of the discussion. However, in the results it is written: "Digital citizenship was mediated by perceived task difficulty…"Digital life was mediated by perceived task difficulty….". These expressions are not aligned with the model.

Another expression: "Each path of mediation was significant for the poor class but not for the non-poor class" is misleading. As is shown in table 3 the moderation effect for X1, X2, X2 à M1 àY shows that it was not significant for the poor. Only the M2 (digital life) and the dual mediation were significant for the poor.

Although the Hayes PROCESS methodology is a very elegant and easy to use, I suggest adding some simplest methodology which can shad lights on the conclusion. As the moderation effect of population (poor vs non-poor) is clear, I suggest running separate multivariable analysis, one for the poor and one for the non-poor to present the predictors of life satisfaction.

I have some minor remarks:

1.   In row 222 it is written: " The participants’ demographic characteristics are listed in Table 1.". However, table 1 presents not only the demographics but also the mean score of life satisfaction. This should be clear in the text.

2.   The title of table 1 is misleading. I suggest using the following: Mean scores of life satisfaction by demographic characteristics.

3.   In table 2 there is a mistake. It presents M±SD of life satisfaction = 0.61±0.61. This should be in the range of the M±SD in table 1.             I suggest adding to table 1 a total population score of life satisfaction M±SD.

Reviewer 3 Report

End of abstract is repetitive; please streamline.

Good topic and model

Add more ideas on future research

Spell out journal titles

Round 2

Reviewer 1 Report

NA

NA